# Effect of Oxygen Plasma Pre-Treatment on the Surface Properties of Si-Modified Cotton Membranes for Oil/Water Separations

**DOI:** 10.3390/ma15238551

**Published:** 2022-11-30

**Authors:** Leila Ghorbani, Daniela Caschera, Babak Shokri

**Affiliations:** 1Laser and Plasma Research Institute, Shahid Beheshti University, Tehran 19839, Iran; 2Institute for the Study of Nanostructured Materials, Strada Provinciale 35 d, n. 9, Montelibretti, 00010 Rome, Italy; 3Faculty of Physics, Shahid Beheshti University, Tehran 19839, Iran

**Keywords:** oxygen plasma pre-treatment, dip coating, plasma polymerization, PDMS, oil/water separation, cotton textile

## Abstract

Hydrophobic and oleophilic Si-based cotton fabrics have recently gained a lot of attention in oil/water separation due to their high efficiency. In this study, we present the effect of O_2_ plasma pre-treatment on the final properties of two Si-based cotton membranes obtained from dip coating and plasma polymerization, using polydimethylsiloxane (PDMS) as starting polymeric precursor. The structural characterizations indicate the presence of Si bond on both the modified cotton surfaces, with an increase of the carbon bond, assuring the success in surface modification. On the other hand, employing O_2_ plasma strongly changes the cotton morphology, inducing specific roughness and affecting the hydrophobicity durability and separation efficiency. In particular, the wettability has been retained after 20 laundry tests at 40 °C and 80 °C, and, for separation efficiency, even after 30 cycles, an improvement in the range of 10–15%, both at room temperature and at 90 °C can be observed. These results clearly demonstrate that O_2_ plasma pre-treatment, an eco-friendly, non-toxic, solvent-free, and one-step method for inducing specific functionalities on surfaces, is very effective in enhancing the oil/water separation properties for Si-based cotton membranes, especially in combination with plasma polymerization procedure for Si-based deposition.

## 1. Introduction

In the last years, the increasing production of oily wastewater from textile, food, leather, Metal processing, oil and gas industries, and frequent requests for effective treatment for oil spill removal have given a lot of importance to the scientific research of oil-water separation, with the aim of reducing the induced severe environmental and economic impacts [1,2,3,4,5,6,7]. In particular, the development of membranes for oil/water separation grasps more attention with respect to other separation methods due to their high efficiency and non-secondary pollution [1,2,4,7,8,9,10]. Several studies have been conducted on oil/water separation membranes by using different materials and physical and chemical methods for developing separation filters. In particular, Si-based materials have been largely studied and employed for membrane functionalization, and several works proposed the combination of materials and procedures to develop oil/water separation membranes. Yeom et al. fabricated a superhydrophobic mesh and sponge through a simple and fast dip-coating of silica nanoparticles (SiNP), which improved the hydrophobicity of the surface [11]. Ma et al. suggested polymerized reaction of silica nanoparticles on electrospun PI nanofibers to fabricate a separation membrane [12]. In another study, Woo et al. fabricated a nanostructured glass microfiber by using the vertical polymerization coating method [13]. Zhang et al. grew nanofilaments on the textile via chemical vapor deposition of trichloromethylsilane for obtaining oil-water separation membranes [14], while in a study by Song et al., the nano-SiO_2_ solution was prepared by sol-gel method and deposited on the cotton textile by a dip-coating approach to obtain superhydrophobic/superoleophilic fabrics [15]. Si-modified membranes present very promising performances in oil/water separation, but these systems often undergo poor adhesion and low durability, especially under drastic work conditions (i.e., high temperatures and long work-cycles [16,17,18,19,20]. In this context, the research of more effective methods to enhance the wettability and oil separation performances of Si-based membranes using green technologies is particularly appealing.

One of the most used materials for the fabrication of Si-based membranes for the oil/water separation process is polydimethylsiloxane (PDMS) due to its intrinsic hydrophobicity [16,17,18,21]. Several methods could be considered for depositing PDMS, such as dip-coating [22], plasma polymerization [23,24], electrospinning [17], vapor deposition [20], and chemical vapor deposition [18].

In this study, we took into consideration two methods, dip-coating, and plasma polymerization, for the fabrication of Si-based coating on cotton, using PDMS as the precursor. We also investigated the influence of an O_2_ plasma pre-treatment on the final properties of the as-produced systems. 

Plasma pre-treatment is one of the most effective methods to modify a surface, introducing specific functional groups [25] and/or changing its roughness [26]. The surface, as modified, changes its intrinsic properties as adhesion capability, wettability, and morphology [27,28]. Furthermore, specific plasma pre-treatment can also influence the properties of the coating that can be deposited on the modified surface due to the combination of imposed roughness and the presence of chemical groups on the substrate surface [29].

In particular, it has been observed that an O_2_ plasma treatment could be particularly effective in influencing the wettability properties of textile-based systems [30]. In this view, it could introduce the fabrication of oil/water membranes procedure [2,31].

This work aims to demonstrate that O_2_ plasma pre-treatment has an effective influence in enhancing the performances of Si-modified cotton as the membrane for oil/water separation. In this view, structural and functional properties of the as-fabricated Si-cotton membranes have been evaluated, with and without a pre-treatment of O_2_ plasma on the cotton substrate. It has already been demonstrated that plasma pre-treatment can significantly affect the hydrophobicity of the surface by modifying the roughness of the surface [32]. As a consequence, our results indicated that O_2_ plasma pre-treatment has a significant impact on the final morphology of Si-coatings, more evident for samples fabricated by the plasma polymerization method. Thanks to the specific hierarchical structure imposed by this pre-treatment, an increase in the contact angle values and separation efficiency of both filters have been observed. In particular, Si-cotton membrane fabricated by using only plasma processes (O_2_ pre-treatment + plasma polymerization) showed an improvement in efficiency up to 15%, and this ability has been retained even after 30 separation cycles. In this contest, a one-step and eco-friendly O_2_ plasma pre-treatment can be considered an influencing factor for the fabrication of oil/water separation filters with very encouraging properties even for applications in extreme conditions and environments. 

## 2. Experiment

### 2.1. Materials

Cotton (from a local store) was cut into 10 cm × 10 cm samples, washed in a 5 wt.% aqueous detergent solution (obtained by diluting in distilled water a proper amount of a commercial clothes soap), and finally dried in an oven at 50 °C for 40 min. To prepare a final homogenous Si precursor solution with low viscosity, PDMS (Dow Corning, Midland, MI, USA, Sylgard186 elastomer base-2g) and PDMS (Sigma Aldrich, St. Louis, MO, USA, Sylgard186 curing agent-0.2 g) were mixed into 50 mL of toluene. All other materials were purchased from Sinopharm Chemical Reagent Co., Ltd., Shanghai, China.

In Figure 1, the polymeric chemical structure of PDMS is reported.

### 2.2. Dip-Coating

The cotton fabric was immersed in the PDMS toluene solution for one minute, squeezed to remove the excess, and then cured at about 125 °C in the oven for 20 min. The membrane obtained by this procedure has been labeled as Si-Cot_Dip. 

### 2.3. Plasma Polymerization

A homemade low-pressure capacitively coupled plasma (PECVD) system was employed for plasma deposition. The cotton textile was placed on the power electrode (14 cm in diameter, RF power 13.56 MHz), connected to an automatic impedance matching network.

Argon was used as a carrier gas for introducing the PDMS-based solution vapor into the plasma chamber. The deposition process was carried out at 20 W RF power for 20 min at 18.6 Pa (working pressure). The PDMS/Ar mixture flow was fixed at 20 sccm. The volume of the plasma chamber was 15 L. This filter was labeled as Si-Cot_Plasma.

### 2.4. O_2_ Plasma Pre-Treatment

O_2_ plasma pre-treatment was performed on cotton substrates, using the same PECVD system, before PDMS deposition. The plasma conditions were set as follows: 20 W RF power, 20 sccm O_2_ gas flow, 10 Pa working pressure, and 30 min treatment time. The process was carried out at 25 °C and checked by a thermocouple. 

Then the as-treated cotton samples were modified with PDMS according to the procedures previously reported (2.2 and 2.3 paragraphs), and the as-fabricated Si-cotton membranes were labeled as follows:

Si-Cot_Dip/O_2_, for the filter, pre-treated with O_2_ plasma and then modified with PDMS by dip-coating.

Si-Cot_Plasma/O_2_, for the filter, pre-treated with O_2_ plasma and then deposited with PDMS by plasma polymerization.

### 2.5. Structural and Morphological Characterization

X-ray photoelectron spectroscopy (XPS) analysis was carried out using a Thermo Scientific KA1066 spectrometer with monochromatic Al Kα (with energy 1486.6 eV) X-ray radiation. The spectra were collected with 0.1 eV step at 50 eV constant pass energy. The calibration of the binding energy (BE) scale was achieved with respect to the Au 4f7/2 peak, set at BE = 84.0 eV. The accuracy of the binding energy (BE) scale was ±0.1 eV. Spectroscopic data were processed by CasaXPS software, subtracting the Shirley background and using a peak-fitting routine with a symmetrical product mix of Gaussian−Lorentzian functions L = 30% and G = 70%.

Fourier-transformed infrared (FTIR) spectra were performed using the Thermo Nicolet model nexus 470 in the range of 700–4000 cm^−1^. 

SEM analysis was collected by a Hitachi, with an accelerating voltage of 15 kV, pre-coating the samples with a gold layer to prevent any charges.

Wettability measurements were performed by the sessile drop method in standard conditions (at atmospheric pressure and room temperature). The cotton samples were fixed on a microscope slide with adhesive tape to ensure the planarity of the surface. A droplet of 4 μL of deionized water was dropped on the surface, and the photos of the drops were taken with a Canon camera and put on the side of the experimental set-up. The reported water contact angles (WCA) were averaged over three measurements in different spots. The contact angles were evaluated using the Image J software (1.52v).

### 2.6. Laundry Test

The durability of the surface properties of the systems, consequently the adhesion of the Si-coating, and the wettability performances of the membranes were evaluated by the laundry test. The Si-coated cotton (10 cm × 10 cm) was immersed in 200 mL distilled water at 40 °C and 80 °C, respectively, and two stirring bars of about 2 cm. It was left under magnetic stirring at an average speed of 600 rpm for 10 min, then dried in an oven at 50 °C. The test was performed at 40 °C and 80 °C. Twenty wash cycles of 10 min each have been performed for each sample, and five WCA measurements were performed for each cycle, reporting the average value [34]. The WCA measurements were repeated 20 times.

### 2.7. Separation Efficiency 

The separation efficiency of the Si-coating textiles was tested at room temperature and 90 °C by using several oils: paraffin (viscosity 13–18 cSt), gasoline (viscosity 0.006 Poise), pump oil (viscosity > 500 cSt), and hexadecane (viscosity 0.0034 Poise), mixed with water (viscosity 1 cSt). In detail, a room temperature mixture and hot oil/hot water mixture with the 1:1 volume ratio were prepared. The hot oil/water systems were firstly mixed at room temperature and then heated at 100 °C, in a water bath, due to the high flammability.

These selected experimental conditions result in being closer to those normally detectable in practical situations, including the dispersion of oil in seawater or industrial wastewater.

The Si-modified cotton was placed on a funnel, and the oily solutions were poured on. Then, after the separation process, the efficiency was calculated by the following equation:$$\mathrm{separation}\;\mathrm{efficiency}\;\%=(V_{1}/V_{0})\times 100$$
where *V*_1_ is the volume of the water collected on the filter after the separation test, and *V*_0_ denotes the starting volume of the initial water [4,35,36]. The tests were repeated 30 times for different oil/water mixtures. The separation efficiency was calculated five times for each cycle, reporting the average values and calculating the corresponding errors, as recommended in [34].

## 3. Results and Discussion

The achievement of the cotton surface functionalization with Si-based coatings has been investigated by studying the chemical structure and composition of the Si-Cot_Dip/O_2_ and Si-Cot_Plasma/O_2_ systems, using XPS and FT-IR ATR measurements. The comparison of the XPS survey of samples (Figure 2a,e,i) shows several changes in the relative intensity of C1s and O1s signals. Furthermore, the presence of new peaks at 101.8 and 152.8 eV, related to Si2p and Si2s [37], on the Si-Cot_Dip/O_2_ and Si-Cot_Plasma/O_2_ surfaces (Figure 2e,i) is clearly observable. This evidence indicates the success of both methods (dip-coating and plasma polymerization) in adding Si-based coatings on the cotton surface. 

In Figure 2b, the C1s spectrum of the pristine sample presents the three typical bands for cotton, at 284.8, 286.5, and 287.7 eV, related to C-C, C-O-C, and C=O bonds, respectively [38]. The intensity of C-C, C-O-C, and C=O after dip-coating (Si-Cot_Dip/O_2_*)* decreased in comparison with pristine cotton (Figure 2f). In contrast, after plasma polymerization (Si-Cot_Plasma/O_2_) (Figure 2j), the C-C signal increases in intensity, while C-O-C and C=O signals are reduced with respect to cotton. This effect is due to the fact that the plasma polymerization process causes the incorporation of carbon species, with respect to oxygen, on the coating. In contrast, the dip-coating method is responsible for creating more Si species and decreasing the oxygen ones. On the other hand, the O1s spectrum demonstrates that the O_2_ amount can be resolved into several components. In Figure 2c, the pristine cotton, the band around 532.6 eV is assigned to C-OH, related to unsaturated bonds on the surface [39]. This signal disappears in Si-Cot_Dip/O_2_ and Si-Cot_Plasma/O_2_ samples (Figure 2g,k), indicating that both methods can obtain unsaturated bonds on the surface passively. Moreover, the signals around 532.6 and 533.4 eV are due to C-O and Si-O-C, respectively [21]. The intensity of the Si-O-C bond increases more in Si-Cot_Dip/O_2_ sample than that in the Si-Cot_Plasma/O_2_ sample [40,41].

Furthermore, the evaluation of the atomic percentages of silicon, oxygen, and carbon (Appendix A) confirms the success of the two methods in depositing a Si-based coating.

Figure 3 shows the ATR-FTIR spectra of pristine cotton, O_2_ plasma pretreated cotton, O_2_ pre-treatment plasma, Si-Cot_Dip/O_2_, and Si-Cot_Plasma/O_2_ samples. In pristine cotton, the absorption peaks related to the O-H bond and the asymmetric stretching vibration C-H are present at 3335 and 2902 cm^−1^, respectively [32]. Furthermore, the peaks at 1054–1026 cm^−1^ are attributed to the C-O-C stretching vibrations. The pristine cotton spectrum is comparable with the literature [32], confirming that no specific impurities are present. FT-IR spectrum of O_2_ plasma pretreated (Figure 3b) cotton is very similar to that of the cotton pristine, but some differences are visible at those frequencies attributed to the C=O vibration groups in the carbonyl structure and O–H stretching vibration, i.e., in the range around 1500–1750 and 2700–3100 cm^−1^, respectively [32]. In particular, the effect of plasma surface oxidation can be related to the appearance of a small peak at 1540 cm^−1^ due to COO–stretching vibrations [42,43,44,45].

The introduction of these specific C-O and O-H groups on the cotton surface, thanks to the O_2_ plasma, leads be beneficial for the improvement of the Si-based coating adhesion since it can give the possibility to create more stable bonds between the coating and the cotton substrate [46,47]. 

The bands at 1007 and 1072 cm^−1^ in both the FT-IR spectra of Si-Cot_Dip/O_2_ and Si-Cot_Plasma/O_2_ samples (Figure 3c,d) are characteristic of Si–O–Si asymmetric and symmetric stretching vibrations [48]. However, the intensity of these bonds is higher in Si-Cot_Dip/O_2_. Also, symmetric rocking at 864 cm^−1^ and stretching at 785 cm^−1^ vibrations in both coating samples can be attributed to the Si–CH_3_ group. Related to Si-Cot_Dip/O_2_ and Si-Cot_Plasma/O_2_, the absorption band at 1410 cm^−1^ is assigned to CH_3_ asymmetric bending in Si–CH_3_ bonds, while the signals at 2965 and 2906 cm^−1^ are the asymmetric stretching and symmetric stretching of the CH_3_ group, respectively [49,50]. Also, the low-intensity band at 1450 cm^−1^ is attributed to the asymmetric bending vibrations of the CH_2_ group in the Si–CH_2_–CH_2_–Si link [49,50]. In addition, the Si–H stretching vibration presents signals at 2100–2300 cm^−1^, while the band at 1629 cm^−1^ is due to the stretching of the C=O [49,50]. The presence of Si-related peaks in the FT-ATR spectra of the Si-Cot_Dip/O_2_ and Si-Cot_Plasma/O_2_ confirms the success of the deposition processes. Moreover, the intensities of the Si–H bonds at 1030, 1050, 2122, 2160, 2179, 2250, and 2333 cm^−1^ for Si-Cot_Dip/O_2_ are more evident than those for Si-Cot_Plasma/O_2_ sample, while C-H bonds at 2895, 3024, 3050, and 3337 cm^−1^ and C-C bonds at 1443 and 1495 cm^−1^ for Si-Cot_Plasma/O_2_ are more intense than those for Si-Cot_Dip/O_2_. These results confirm the observation of XPS measurements about the percentage of Si and C in the samples fabricated by two dip-coating and plasma polymerization techniques.

Although the plasma pre-treatments have almost no influence on the chemical composition of the coatings [51], it is well known that O_2_ plasma pre-treatments can strongly modify the surface of the substrate, inducing, in the following deposited coatings, specific hierarchical growth and opportune morphologies, and contributing to the final surface properties [52,53]. To assess the influence of plasma pre-treatment, SEM analysis has been carried out, and the corresponding images, before and after O_2_ plasma, for both dip-coating and plasma polymerization methods are shown in Figure 4. Figure 4a shows that pristine cotton fibers present a relatively smooth surface. When the cotton is covered with Si-based coatings by dip-coating and plasma polymerization methods, differences in morphologies could be visible. For the dip-coating method, the surface of the Si-Cot_Dip appears more monotonous and homogeneously covered, and no recognizable structures are visible (Figure 4b). On the contrary, the coating obtained from plasma polymerization, Si-Cot_Plasma, (Figure 4c), presents a rougher surface associated with visible nanostructures on the fiber. In Figure 4d, the effect of 30 min of O_2_ plasma treatment on the cotton surface is presented: the roughness and the morphology of the fibers result in being significantly altered, and many holes are created on the surface due to O_2_ plasma treatment. The influence of oxygen pre-treatment on the coating morphology and growth is vividly recognized in Figure 4e,f. According to Figure 4e, the Si-coating obtained by the dip method (Si-Cot_Dip/O_2_) retains most of its morphological features, showing a compact and homogeneous coverage of about 1 µm in thickness, even if some effects of the O_2_ plasma pre-treatment start to be visible on the Si growth. Furthermore, in Figure 4f, Si-Cot_Plasma/O_2_ coating displays a more irregular structure, created on the surface fibers, with respect to the Si-Cot_Plasma coating obtained without a pre-treatment of oxygen (Figure 4c), permitting the growth of columnar nanostructures with a size of ≈1 µm. 

According to the Cassie–Baxter’s theory [54], the generation of micro and nanostructures on the surface can increase the hydrophobicity thanks to the air pockets that can be trapped in these structures [53]. Previous research has also shown that O_2_ plasma pre-treatment on textiles could strongly affect the adhesion surface [55], making the following Si-based coating more adhesive on cotton. 

Figure 5 shows the macroscopic hydrophobicity and oleophilicity of both Si-Cot_Dip/O_2_ and Si-Cot_Plasma/O_2_. The contact angle (CA) results are summarized in Table 1, in which cotton pristine is added for comparison.

Si-coating samples without O_2_ plasma pre-treatment present quite similar WCA values and great hydrophobic behavior. The presence of Si bonds and the changing of the surface morphology are conditions for final higher WCA values [56,57]. Additionally, increasing the amount of carbon on the cotton surface after coating, as previously observed by XPS, could be an important issue for predicting an increase in surface hydrophobicity [58]. More interestingly, O_2_ plasma treatment has a positive influence by increasing the WCAs, and at the same time, by decreasing the sliding angle values for both Si-Cot_Dip/O_2_ and Si-Cot_Plasma/O_2_. In particular, Si-Cot_Plasma/O_2_ shows the most appropriate combination of WCA and sliding angle for having a hydrophobic surface that should be used as a filter. The micro and nano hierarchical structures of Si-Cot_Plasma/O_2_, similar to the surface of blue lotus [59], ensure the suitable wettability behavior. In addition, the low sliding angle of Si-Cot_Plasma/O_2_ permits its use not only as a vertical filter but in a wide range of angles (15–90°) since it can repel water even at angles of 15°, very close to the horizontal position.

Laundering tests permit the evaluation of the durability of Si-based coatings on cotton. Figure 6 shows the variation of the contact angle of Si-Cot_Dip, Si-Cot_Dip/O_2_, Si-Cot_Plasma, and Si-Cot_Plasma/O_2_ after laundering at 40 °C and 80 °C. It is evident that after 20 laundry cycles, the WCA values for the Si-Cot_Dip and Si-Cot_Plasma (Figure 6a,c) decrease at about 125–128°, independently from the laundering temperature. It is noteworthy that the O_2_ plasma pre-treatment results in an increase in the starting WCA values of about 10° and 15° for Si-Cot_Dip/O_2_ and Si-Cot_Plasma/O_2,_ respectively. Furthermore, the contact angle remains quite constant in the range 135–140° for both Si-Cot_Dip/O_2_ and Si-Cot_Plasma/O_2_, even after twenty laundering cycles at 40 °C and 80 °C (Figure 6b,d). This behavior indicates that O_2_ plasma pre-treatment not only can increase the hydrophobicity of the filter but can also have a great influence on the adhesion of the Si-based coatings. The effect is to obtain filters with stable and higher hydrophobicity durability. In addition, it is well known that the generation of specific hierarchical nanostructures on the cotton can also affect the oleophilicity of the fiber as an effect of capillary [60].

The separation efficiency of the Si-Cot_Dip and Si-Cot_Dip/O_2_, Si-Cot_Plasma and Si-Cot_Plasma/O_2_ as oil/water filters is evaluated up to 30 separation cycles, at room temperature (25 °C) and 90 °C, for mixtures of oils/water (paraffin, gasoline, pump oil, and hexadecane) (Figure 7 and Figure 8). It is evident that the separation efficiency of Si-Cot_Dip for mixture at 25 °C (Figure 7a) is retained even up to about 85% after 30 tests. This separation efficiency after using O_2_ pre-treatment increases by about 10%, reaching approximately 95% at room temperature (Figure 7b and Figure 8b).

The separation efficiency for Si-Cot_Plasma is about 80–75% even after 30 cycles (Figure 7c and Figure 8c). The O_2_ pre-treatment, instead, permits an increase in separation efficiency of about 15%, efficiency reaching approximately 90% even at higher temperatures (Figure 7d and Figure 8d). The slight efficiency increase for Si-Cot_Plasma/O_2_ (15%) with respect to the Si-Cot_Dip/O_2_ (10%) could be attributed to the different final morphologies, as observed from SEM analysis (Figure 4e, f): the combination of the two plasma processes (O_2_ pre-treatment and plasma polymerization) induces a rougher and columnar growth of the Si-based coating on cotton. This particular structure has been demonstrated to be more effective in oil/water separation procedures [58]. Nevertheless, the observed soaring properties also for Si-Cot_Dip/O_2_ confirm the positive effect of O_2_ plasma pre-treatment on the development of a separation membrane system with higher separation efficiency, even in drastic conditions, without the need for any auxiliary force.

To prove the promising performances of the O_2_ plasma pre-treatment as an effective way to increase separation efficiency even for the hot oil/water mixture, a comparison with other filtering cotton systems is provided in Table 2. It is evident that the Si-based cotton filter proposed in this work possesses identical or even superior properties at room temperature with respect to other filters. Furthermore, very few investigations have been carried out on the reusability and separation efficiency of filters in the presence of high-temperature mixtures. When higher filtration temperatures are considered, the analysis of Table 2 highlights that most of the materials proposed in the literature present low efficiency, or, if the performances are acceptable, usually the starting materials are not eco-friendly, or they are prepared using complex or expensive methods. On the contrary, the Si-based cotton filters show suitable performance in separation efficiency at high temperatures. Furthermore, the proposed synthetic methods are greener than traditional approaches. In addition, O_2_ plasma pre-treatment that increases the efficiency and durability of the filter is a facile and eco-friendly method. These considerations make the O_2_ plasma pre-treatment a very appealing and suitable method for increasing the efficiency of oil/water separation at different temperatures.

## 4. Conclusions

In this paper, we explored the effect of O_2_ plasma pre-treatment on the oil/water separation properties of Si-based cotton filters. The modified membranes have been realized using different methods, a dip-coating procedure, and plasma polymerization, using the eco-friendly PDMS as Si precursor. Structural characterizations have been carried out to assess the success of the Si functionalization by XPS and FT-IR spectroscopies. Furthermore, a deep analysis of the morphologies of the membranes has been evaluated, taking into account the effect of the plasma pre-treatment on the roughness and surface structures. SEM highlighted that O_2_ plasma pre-treatment is very effective in inducing the growth of a specific hierarchical structure on the cotton surface. This feature is partially retained even in the membrane functionalized by the dip-coating method. As expected, both Si-Cot_Dip and Si-Cot_Plasma show very promising performances in oil/water separation tests, but their capability and durability result in an enhancement when the Si deposition is preceded by O_2_ plasma pre-treatment. These results clearly point out the positive effect of O_2_ plasma pre-treatment on the hydrophobic/oleophilic properties of the Si-based cotton membrane. The possibility of the plasma process to opportunely create and modulate a hierarchical structure on the fiber surface, combined with the Si-based coating properties, permits an increase in the separation efficiency of the membrane, also imparting great durability even after several cycles of usage and laundering. In particular, the combination of plasma processes (O_2_ pre-treatment + plasma polymerization) results in the most effective way of developing oil/water separation filters, opportunely controlling the surface properties of the membranes.

## Figures and Tables

**Figure 1 materials-15-08551-f001:**
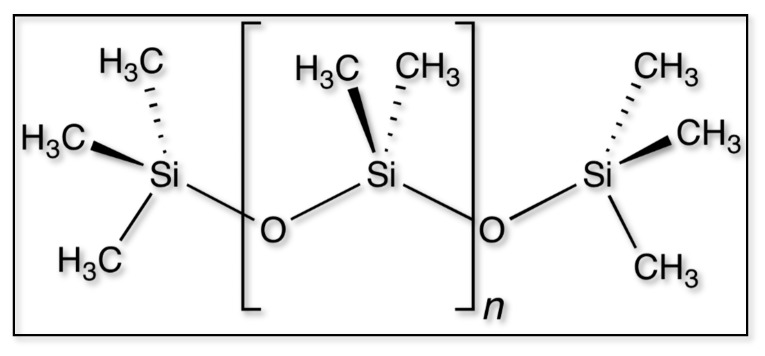
PDMS polymeric structure [33].

**Figure 2 materials-15-08551-f002:**
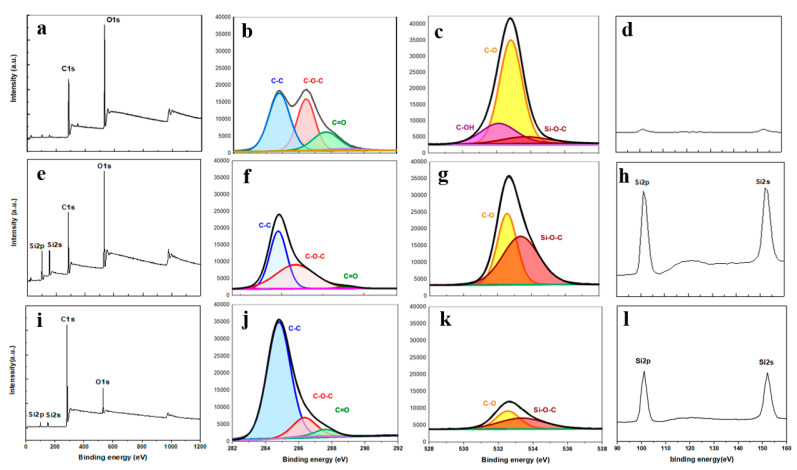
XPS measurements for pristine cotton (**a**) survey spectrum, (**b**) C1s spectra, (**c**) O1s spectra, (**d**) Si spectra; Si-Cot_Dip/O_2_ (**e**) survey spectrum, (**f**) C1s spectra, (**g**) O1s spectra, (**h**) Si spectra; Si-Cot_Plasma/O_2_ (**i**) survey spectrum, (**j**) C1s spectra (**k**) O1s spectra (**l**) Si spectra.

**Figure 3 materials-15-08551-f003:**
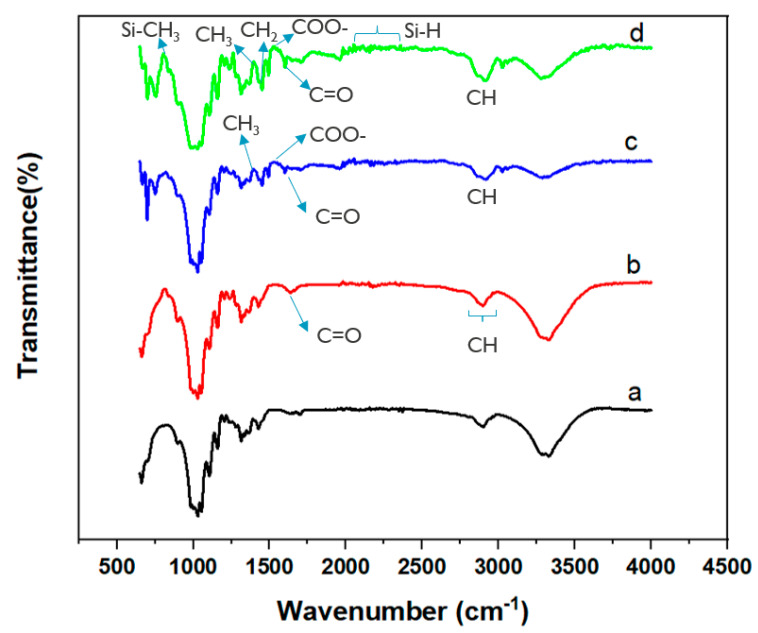
ATR-FTIR spectroscopy of (a) pristine cotton, (b) O_2_ plasma pre-treatment, (c) Si-Cot_Dip/O_2_, (d) Si-Cot_Plasma/O_2_ samples.

**Figure 4 materials-15-08551-f004:**
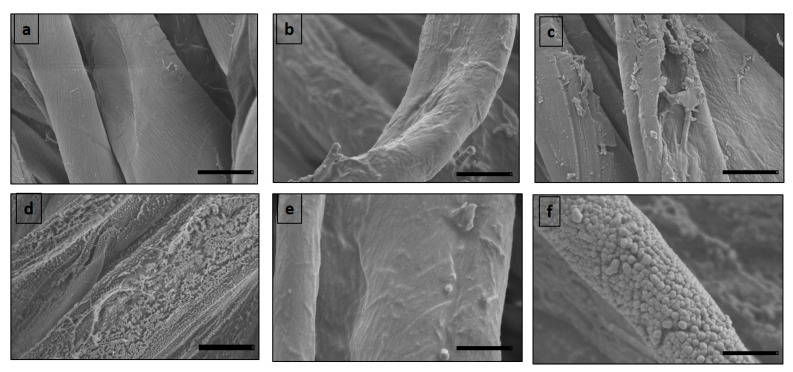
SEM images of (**a**) Pristine cotton, (**b**) Si-Cot_Dip, (**c**) Si-Cot_Plasma, (**d**) Cotton after oxygen plasma pre-treatment, (**e**) Si-Cot_Dip/O_2_, (**f**) Si-Cot_Plasma/O_2_ (Scale bar: 10 µm).

**Figure 5 materials-15-08551-f005:**
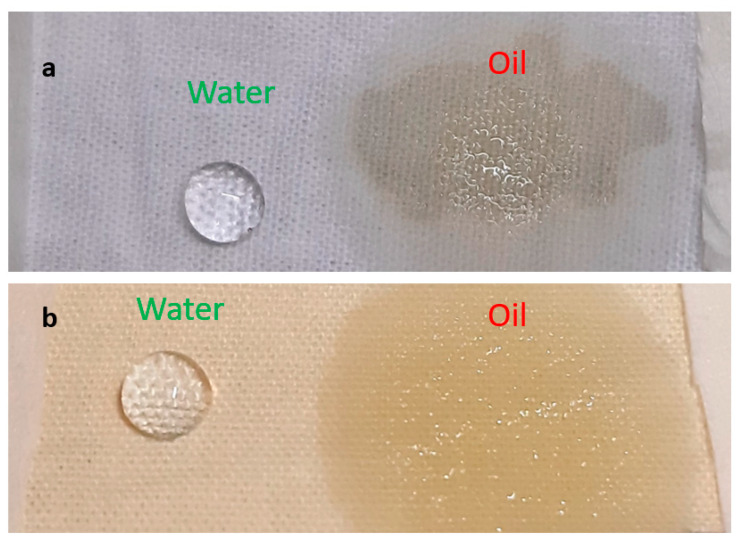
Water and oil on the surface of (**a**) Si-Cot_Dip/O_2_, (**b**) Si-Cot_Plasma/O_2_.

**Figure 6 materials-15-08551-f006:**
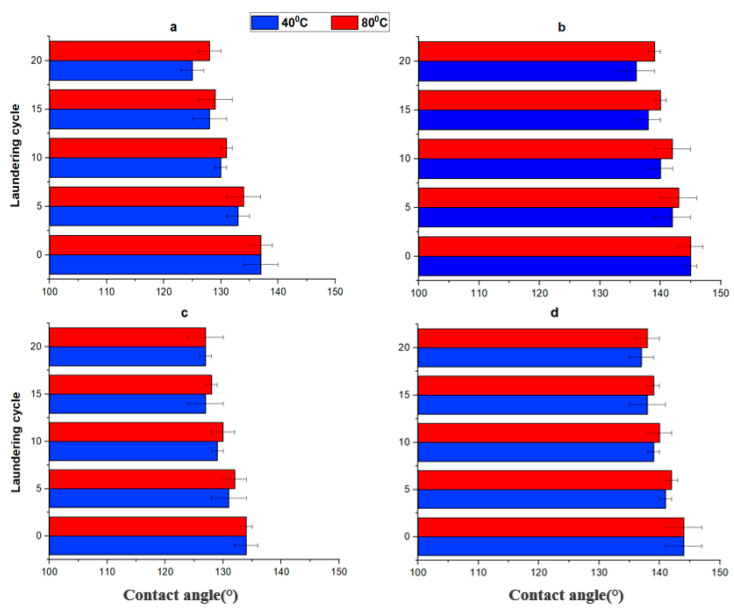
Variation of WCA of (**a**) Si-Cot_Dip and (**b**) Si-Cot_Dip/O_2_ (**c**) Si-Cot_Plasma (**d**) Si-Cot_Plasma/O_2_ after laundering tests at different temperatures.

**Figure 7 materials-15-08551-f007:**
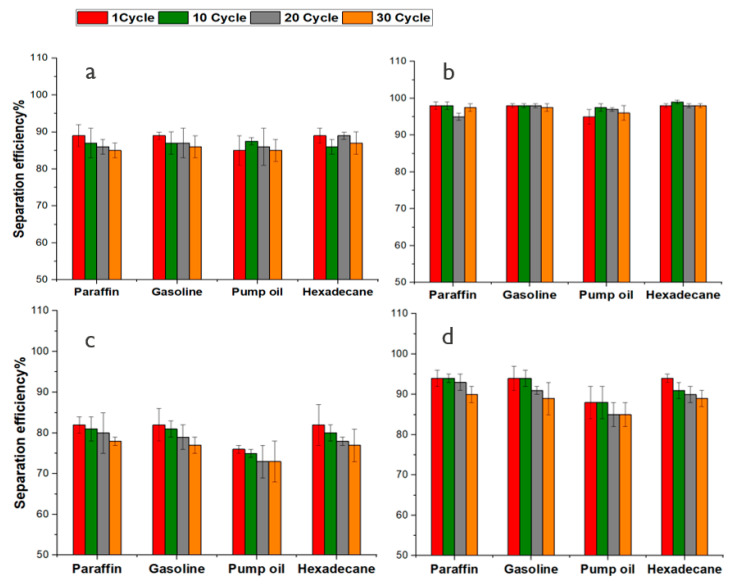
The separation efficiency of (**a**) Si-Cot_Dip (**b**) Si-Cot_Dip/O_2_ (**c**) Si-Cot_Plasma (**d**) Si-Cot_Plasma/O_2_ at room temperature for oil/water mixtures.

**Figure 8 materials-15-08551-f008:**
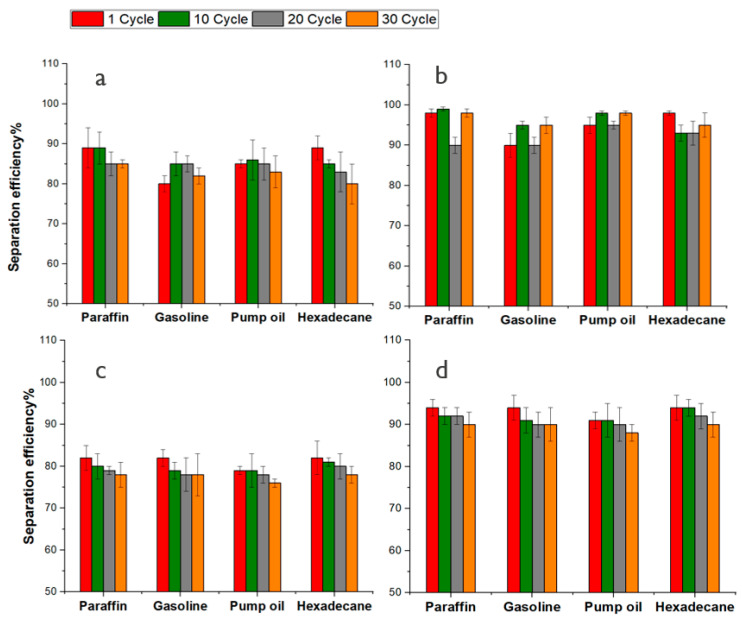
The separation efficiency of (**a**) Si-Cot_Dip (**b**) Si-Cot_Dip/O_2_ (**c**) Si-Cot_Plasma (**d**) Si-Cot_Plasma/O_2_ at 90 °C for several oil/water mixtures, after different cycles.

**Table 1 materials-15-08551-t001:** WCA and Sliding angle measurements for pristine cotton and Si-coating samples.

Sample	Water Contact Angle (°)	Sliding Angle (°)
Pristine cotton	0	-
Si-Cot_Dip	137 ± 3	50 ± 2
Si-Cot_Dip/O_2_	145 ± 3	30 ± 2
Si-Cot_Plasma	134 ± 3	25 ± 2
Si-Cot_Plasma/O_2_	144 ± 3	15 ± 2

**Table 2 materials-15-08551-t002:** Separation efficiency (SE) of different materials used as filters for hot oil/water mixture.

Material	Method	Type of Oils	SE of Room Temperature Mixture	Number of Cycles	SE of High Temperature Mixture	Number of Cycles	Reference
PDMS	Plasma polymerization	Gasoline Hexadecane	90	30	90	30 (Hot oil/Hot water)	This work
PDMS	Dip coating	Gasoline Hexadecane	95	30	95	30 (Hot oil/Hot water)	This work
DLC	Coating by plasma	Pump oil	100	1	-	-	[58]
Acrylamide and acrylonitrile	Graft copolymerization	Crude Olive Diesel	95–99	1	-	-	[61]
HMDSO	Plasma polymer	Oil Hexane	99	1	-	-	[62]
HDTMS and SA	Dip coating	Hexadecane Diesel	100	1	-	-	[63]
HDTS	Dip coating	chloroform	98	10	-	-	[64]
PDMS	Vapor deposition	Hexane	99	1	-	-	[20]
PDA-Ca complex	Dip coating	Gasoline	-	-	97	80 (Hot water)	[65]
LiCl/DMAc	Dip coating	hexane	98.5	60	-	-	[66]
PEDOT-PSS hydrogel	Chemical polymerization	Diesel	-	-	99	50 (Hot water)	[67]
Mg (NO_3_)_2_	Dip coating	Diesel	10	96	-	-	[68]
Stearoyl	Dip coating	octan	10	100	-	-	[69]
PDMS-Fe3O4@MF	Co-deposition	toluene	30	99.9	-	-	[36]

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
