# Peer review of "Effect of Oxygen Plasma Pre-Treatment on the Surface Properties of Si-Modified Cotton Membranes for Oil/Water Separations"

_materials, 2022, doi:10.3390/ma15238551_

Round 1

Reviewer 1 Report

The authors present the result of O2 plasma pre-treatment on Si-modified cotton membranes for separator production. The reviewer would ask the question and add comments to the manuscript below:

Chap.1 - the Introduction lacks more insight and references to plasma processing or treatment of PDMS with a focus on oxygen plasma

Chap.2 - the samples and methods are not well described. At least the information in the list below has to be added:

2.1 - cotton properties (manufacturer, cotton ratio in fabric, weight/m2, fabric density or a photograph under the microscope); detergent identification (type or composition); is pristine cotton referred for the washed sample, or can detergent change its chem. properties - was it tested by XPS and FTIR?

2.3 - the volume of the low-pressure chamber used for the PECVD process

Line158 - hydrophilic surface does not repeal water ---> It is not clear why authors chose the weight of water instead of oil measuring.

L186-191 - Does plasma polymerization incorporate more C to the surface (detected as a C-C bond), and from what source and why. Can it be related to the results in O1s spectra and figure S1? Alternatively, may it be caused by chain scission and crosslinking of polymer on the surface? 

206-241 - In the FTIR result description, it is unclear how the authors dealt with the "non-flat surface of fabric" in spectra evaluation. How many spectra were measured and averaged for one sample? Moreover, the comparison of peak intensity should be confronted with some "stable peak" typical for cotton textiles (e.g. ref.26). 

L261-269 - In a comparison of fig. 4b with 4e, the O2 plasma pre-treatment has no significant effect on the final PDMS coating. Thus, the presented micrographs do not well-founded the statement about the impact of O2 plasma pre-treatment on Si growth. Moreover, in the case of fig 4f, the statement about the generation of micro/nanostructures is also spurious. The structures in fig. 4f resemble structures in fig. 4c caused by O2 plasma (plasma etching/sputtering common for O2 plasma).

L274 - wording "increasing the adhesion surface" is unclear - does it mean increasing the adhesion of the surface or the surface able to adhere to another substance/material?

 Fig.5 - How was the CA experiment performed? Was the camera from above (as in the example) or a side? How did the authors ensure the planarity of the cotton sample to measure the CA after wetting the surface (wrinkling)? Please specify and add to chap. 2.5

Fig.6 - Can authors comment on why the lower temperature of the laundering cycle causes more profound degradation (lowering of CA) in cases 6a and 6b?

Fig.7-8 - The reviewer suggests changing the figures and joining the results for all samples into fig. with room temperature and fig. with 90°C for better comparison. 

L341 - Was CA and separation efficiency also tested for O2 plasma pre-treated cotton? Does the morphology ensure the main part of the hydrophobic/separation effect?

Author Response

Reviewer 1

The authors present the result of O2 plasma pre-treatment on Si-modified cotton membranes for separator production. The reviewer would ask the question and add comments to the manuscript below:

Chap.1 - the Introduction lacks more insight and references to plasma processing or treatment of PDMS with a focus on oxygen plasma

R: Thanks for this observation. A paragraph about plasma processing and in particular the effect of O2 pre-treatments has been added. Even, more references have been included in the revised version of the manuscript.

Chap.2 - the samples and methods are not well described. At least the information in the list below has to be added:

2.1 - cotton properties (manufacturer, cotton ratio in fabric, weight/m2, fabric density, or a photograph under the microscope); detergent identification (type or composition); is pristine cotton referred for the washed sample, or can detergent change its chem. properties - was it tested by XPS and FTIR?

R: Cotton has been bought in a local market, so much technical information regarding the fabrication process and manufacturer is lacking. The XPS and FT-IR analysis on the pristine cotton is carried out, before and after the use of detergent solution (commercial clothes soap), to assess that no residuals of detergent are present on the surface. Since no differences are observed, the washed cotton has been used and referred to in the text as “pristine”. Furthermore, the comparison with literature data confirmed that the cotton structure is very similar to others already reported and used as “pristine” in these kinds of studies.

2.3 - the volume of the low-pressure chamber used for the PECVD process

R: The volume of the PECVD chamber has been added

Line158 - hydrophilic surface does not repeal water ---> It is not clear why authors chose the weight of water instead of oil measuring.

R: We thank a lot the reviewer for this comment. He/she is right about the repellency of water on a hydrophilic surface. The sentence has been deleted. Since the modified cotton is highly hydrophobic and superoleophobic, during the separation process is more probable that some amounts of oils could be trapped in the cotton structure. For this reason, we decided to use the weight of the water, not passed through the membrane, for a more sensitive evaluation of the efficiency.

L186-191 - Does plasma polymerization incorporate more C to the surface (detected as a C-C bond), and from what source and why. Can it be related to the results in O1s spectra and figure S1? Alternatively, may it be caused by chain scission and crosslinking of polymer on the surface?

R: We thanks very much the reviewer for this observation. The differences in C and O amounts in dip and plasma coating depend on the deposition method. In the Dip coating method, no strong chemical arrangements of the starting polymeric structure are expected and differences could be due only to the curing step, in which maybe some water molecules could be involved. But in % amount, we expected that C, O, and Si amounts will be very similar to PDMS.

On the contrary, the plasma polymerization method is based on a complete rearrangement of the chemical structure of the starting material. The organic parts of the PDMS are expected to react on the cotton surface, through chain scission and crosslinking creating a hydrogenated carbon-based structure in which Si-O-C polymeric structure can be included. For this reason, in DIP coating the final structure of the Si-based coating is more similar to the starting PDMS, while in plasma polymerization the final coating is considered as an amorphous layer of Si-O-C-H polymeric structure.

206-241 - In the FTIR result description, it is unclear how the authors dealt with the "non-flat surface of fabric" in spectra evaluation. How many spectra were measured and averaged for one sample? Moreover, the comparison of peak intensity should be confronted with some "stable peak" typical for cotton textiles (e.g. ref.26).

R: Thank the referees for this observation. In the ATR measurements, the effect of the “non-flat surface” of the cotton on the punctual analysis of the surface structure is negligible. Nevertheless, in order to minimize errors in measurements, the analysis has been repeated on each sample, with the ATR, in five different points and the observed spectra resulted be comparable both in intensity and in structure.

Furthermore, since the polymeric nature of the materials and considering that a quantitative and more accurate evaluation of the coating structure has been already carried out by XPS, we decided for presenting a more qualitative comparison of ATR-FT/IR spectra, just to evidence the presence of differences in particular bands, and to assess the success of the deposition.

L261-269 - In a comparison of fig. 4b with 4e, the O2 plasma pre-treatment has no significant effect on the final PDMS coating. Thus, the presented micrographs do not well-founded the statement about the impact of O2 plasma pre-treatment on Si growth. Moreover, in the case of fig 4f, the statement about the generation of micro/nanostructures is also spurious. The structures in fig. 4f resemble structures in fig. 4c caused by O2 plasma (plasma etching/sputtering common for O2 plasma).

R: Thanks to the reviewer for this observation. It is well known that a plasma pre-treatment could have great effort in modifying the cotton structure, in particular, O2 could introduce specific morphological roughness, due to the etching or sputtering process.

Furthermore, plasma polymerization methods are capable to deposit thin layers, respecting and/or emphasizing the surface structure and morphology.

On the contrary, dip-coating methods can deposit thicker layers that cover uniformly the surface. In this case, the influence of the surface structure on the final coating morphology is negligible.

We opportunely changed the magnification in figure 4, in order to better explain the effect of the O2 plasma treatment on Si-based coating deposition.

L274 - wording "increasing the adhesion surface" is unclear - does it mean increasing the adhesion of the surface or the surface able to adhere to another substance/material?

R: Thanks to the referee for this comment. We intended that the possibility to create a more roughness surface on cotton, thanks to the O2 plasma pre-treatment, permits a better adhesion of the Si-based coating on the cotton surface itself. This is also confirmed by the WCA measurements after the laundry tests, in which the retainment of the hydrophobic properties could be related to the good adhesion of the coating on the cotton. The sentence has been modified according to the referee’s comment.

 Fig.5 - How was the CA experiment performed? Was the camera from above (as in the example) or a side? How did the authors ensure the planarity of the cotton sample to measure the CA after wetting the surface (wrinkling)? Please specify and add to chap. 2.5

R: Thanks to the referee for his/her observation. In WCA measurements, the camera was put on the side of the experimental set-up, while the cotton sample was fixed on the microscope slide with adhesive tape, to assure the planarity of the surface. The WCA angle evaluation was carried out by Image J software. We also add this information to chap 2.5

Fig.6 - Can authors comment on why the lower temperature of the laundering cycle causes more profound degradation (lowering of CA) in cases 6a and 6b?

R: Thanks to the reviewer for this comment. Observing the WCA decrease after different laundering cycles, the values are always above 120°, assuring the superhydrophobicity of the systems is retained, the coatings are well adherent to the cotton surface and the oil/water separation efficiency could be preserved. Nevertheless, we observed that for the samples treated at a lower temperature (40 °C) the decrease of WCA is higher with respect to the high temperature (90 °C) treatment, even the difference, in percentage, is not so relevant. This effect is more evident in the Si-coating obtained by PDMS dip-coated on cotton and it could be due to the inner structure of the Si-coating, more similar to PDMS precursor, that has different behviour at different temperatures. This effect could be quite interesting, and maybe could be improved in future works.

Fig.7-8 - The reviewer suggests changing the figures and joining the results for all samples into fig. with room temperature and fig. with 90°C for better comparison.

R: Thanks to the reviewer for this observation. Figures 7 and 8 have been changed and the results commented consequently.

L341 - Was CA and separation efficiency also tested for O2 plasma pre-treated cotton? Does the morphology ensure the main part of the hydrophobic/separation effect?

R: It is known that O2-plasma pre-treatment changes the morphology of the cotton surface and can induce the formation of OH bonds on the surface making cotton more hydrophilic (see ref. 48. For this reason static WCA was not measurable for O2 plasma-treated cotton and also the separation efficiency. For this reason, the WCA and the separation efficiency tests are carried out only on Si-modified cotton and the results are compared.

Reviewer 2 Report

author shoud also explain about mechanism reaction in using plasma in pretreatment and polymerization.

Author Response

author shoud also explain about mechanism reaction in using plasma in pretreatment and polymerization.

R: Thanks to the referee for his comment. The mechanism of plasma polymerization of PDMS is well known, as reported in ref 23, 24, and cited wherein. Furthermore, we add a paragraph in the introduction in which the plasma pre-treatment mechanism, and in particular the effects of O2 plasma pre-treatments on the cotton surface, is explained. These mechanisms are very well further investigated in literature (ref. 27-32 and others in the paper).

Reviewer 3 Report

Current manuscript describes the effect of O2 plasma pre-treatment on the oil/water separation properties of Si-based cotton filters. After assessing the whole manuscript, some new information can be found. However, some questions should be addressed to improve the manuscript.
My opinions and comments about this manuscript are as follows:

1. The size and border line of each combination graph in Figure 2 are inconsistent.Such as Figure e has no left scale but Figure a and i have.The border lines and ruler is not clear in some pictures. Please make unified adjustment.

2. The XPS figure pristine cotton also exit weak peaks at in binding energies of 101.8 and 152.8 eV, which related to Si2p and Si2swhyThe XPS analysis shows the intensity of the Si-O-C bond significantly enhanced in Si-Cot_Dip/O2 than that of Si-Cot_Plasma/O2 . Why they nearly have the same water contact angle .

3. It was described the FT-IR spectra test in the range of 500-4000 cm1but the Figure 3 shown the wavenumber range of 700-4000 cm1.Please explain.

4. The original SEM image already has a ruler in Figure 4Each scale should be 2μm, not 20μm.It is recommended that the images be more compact, without adding a yellow border

5. There are irregularities in the annotation of references in the paper.Such as [16]-[19] [24][25] [39],[40]....The references cited after the paper were not uniform and standardSome magazines don't have a shorthandThere is no agreement on whether italics are required and Some references have no DOI numbers....,Please check the full text and correct it.

6. The unit of ‘℃’ are not unified in the full textPlease check the full text and correct it. A space between the number and the unit is recommended.

7. The equation of separation efficiency should add %

8. There are some problems in the upper and lower Angle marking. 02 in line 180, cm1 in line 234, Mg(NO3)2 in Table 2.

9. English expression should be improved

Author Response

REVIEWER 3:

Current manuscript describes the effect of O2 plasma pre-treatment on the oil/water separation properties of Si-based cotton filters. After assessing the whole manuscript, some new information can be found. However, some questions should be addressed to improve the manuscript.
My opinions and comments about this manuscript are as follows:

  1. The size and border line of each combination graph in Figure 2 are inconsistent.Such as Figure e has no left scale but Figure a and i have.The border lines and ruler is not clear in some pictures. Please make unified adjustment.

R: the figure has been changed according to the referee’s suggestion

  1. The XPS figure pristine cotton also exit weak peaks at in binding energies of 101.8 and 152.8 eV, which related to Si2p and Si2s,why?The XPS analysis shows the intensity of the Si-O-C bond significantly enhanced in Si-Cot_Dip/O2than that of Si-Cot_Plasma/O2 . Why they nearly have the same water contact angle .

R: Thanks the referee for this comment. Analyzing the elemental composition of pristine cotton, results in a small amount of Si, around 3%, maybe due to some environmental contamination. After deposition by Dip or plasma method, the amount of Si is visibly higher by the XPS, assessing the real deposition of a Si-based coating on cotton.

The wettability properties of a surface are strictly dependent on two factors: the chemical structure and the surface roughness. For the Si-Cot_Dip/O2, the factor that influences the WCA is the chemical structure, which, considering the specific deposition method, is more similar to the PDMS precursor. On the contrary, for Si-Cot_Plasma/O2 the ability of plasma processing in creating a specific hierarchical structure, due to the induced roughness, contributes to imparting the superhydrophobicity. Both systems presented, for different reasons, high (and very close) WCA, but the results demonstrated that the combination O2 pre-treatment + plasma deposition is more effective in obtaining a filter with higher efficiency even at higher temperatures.

  1. It was described the FT-IR spectra test in the range of 500-4000 cm−1,but the Figure 3 shown the wavenumber range of 700-4000 cm−1.Please explain.

R: Thanks to the referee for the observation. We collected the ATR-FT/IR spectra in the range 700-4000 cm−1. For a mistake, in the measurement paragraph, it has been indicated a wrong range. We changed it in the revised text.

  1. The original SEM image already has a ruler in Figure 4,Each scale should be 2μm, not 20μm.It is recommended that the images be more compact, without adding a yellow border

R: The scale in fig.4 has been checked and modified according to the referee’s suggestion

  1. There are irregularities in the annotation of references in the paper.Such as ‘[16]-[19]’ ‘[24][25]’ ‘[39],[40]’....The references cited after the paper were not uniform and standard,Some magazines don't have a shorthand,There is no agreement on whether italics are required and Some references have no DOI numbers....,Please check the full text and correct it.

R: According to the referee’s suggestion, the references have been checked and changed

  1. The unit of ‘℃’ are not unified in the full text,Please check the full text and correct it. A space between the number and the unit is recommended.

R: We checked and modified the “° C” unit in the full text, according to the referee’s suggestion

  1. The equation of separation efficiency should add ‘%’

R: We added the % in the separation efficiency equation, according to the referee’s suggestion

  1. There are some problems in the upper and lower Angle marking. ‘02’ in line 180, ‘cm1’in line 234,’ ‘Mg(NO3)2’ in Table 2.

R: We checked and made the opportune corrections in the full text, according to the referee’s suggestions

  1. English expression should be improved.

R: English has been completely revised

Reviewer 4 Report

The present work studies the effect of oxygen plasma pretreatment on the surface properties of Si-modified cotton membranes for oil/water separation. The article is very relevant to the area and is on the frontier of applied technology in the textile industry. It is also important to emphasize that oil/water separation is fundamental in the modern world, where drinking water has become scarce in many continents and countries. In addition to the oxygen plasma treatment, the authors used Si deposition on the textile in two ways: dip coating and plasma polymerization. Overall, the work is well-written and well-designed. However, some improvements are needed before publication.

1) Materials and Methods need significant improvement. A question that needs to be answered and added to the text is: What temperature was used in plasma polymerization?

Is PDMS preheated before deposition?

Is the methodology used in the present work viable for industrial application?

2) Results and Discussion: What is the thickness of the Si-Cot-Plasma/O2 and Si-Cot-Dip/O2 layers? The answer to this question is paramount in the XPS and FTIR discussions.

3) In SEM images, it is essential to add other magnifications for a better follow-up of what is happening in the treated fabric threads.

I believe the article can be published after these improvements and responses. For now, I suggest major revisions.

Author Response

REVIEWER 4

The present work studies the effect of oxygen plasma pretreatment on the surface properties of Si-modified cotton membranes for oil/water separation. The article is very relevant to the area and is on the frontier of applied technology in the textile industry. It is also important to emphasize that oil/water separation is fundamental in the modern world, where drinking water has become scarce in many continents and countries. In addition to the oxygen plasma treatment, the authors used Si deposition on the textile in two ways: dip coating and plasma polymerization. Overall, the work is well-written and well-designed. However, some improvements are needed before publication.

  • Materials and Methods need significant improvement. A question that needs to be answered and added to the text is:

What temperature was used in plasma polymerization?

R: Thanks to the reviewer for his/her observation. The plasma deposition process was carried out without supplementary heating. We can estimate a plasma processing temperature at the ground electrode (on which the samples are positioned) of about 30-40 °C, according to the process parameters.

Is PDMS preheated before deposition?

R: Before the deposition, PDMS wasn't heated, but just mixed with toluene to obtain the right viscosity for the plasma process.

Is the methodology used in the present work viable for industrial application?

R: Plasma is considered a new green technology that can easily be integrated into the industrial process. Furthermore, no solvents or very low amounts are needed for plasma deposition, reducing indicatively the cost for transport, storage, and disposal of toxic or low-environmentally friendly materials. In this view, the proposed method for fabricating high-efficiency cotton-based filtering membranes is particularly suitable and effective for industrial applications.

2) Results and Discussion: What is the thickness of the Si-Cot-Plasma/O2 and Si-Cot-Dip/O2 layers? The answer to this question is paramount in the XPS and FTIR discussions.

R: Thanks to the reviewer for this comment. The deposition parameters have been opportunely tuned in order to obtain coatings with comparable thickness. In our case, both coatings have a thickness of about 1µm.

3) In SEM images, it is essential to add other magnifications for a better follow-up of what is happening in the treated fabric threads.

R: According to the referee’s observation, we changed the SEM images to better evidence the differences in morphology for the treated fabrics.

I believe the article can be published after these improvements and responses. For now, I suggest major revisions.

Round 2

Reviewer 1 Report

The authors present the result of O2 plasma pre-treatment on Si-modified cotton membranes for separator production. Some of the proposed revisions were well made, and the authors answered the reviewer's questions adequately. The manuscript can be accepted in revised form after some minor changes:

The reviewer suggests changing the range of the Y-axis in figure 8c to match the other presented results of separation efficiency.

Furthermore, the authors should check the spelling and grammar in added text.

Author Response

We thank the reviewers and the editor for the time and effort that they invested into the review of our manuscript, and for their helpful comments and suggestions.

Reviewer 1:

We thank a lot the Reviewer for his/her effort in helping us in improving our manuscript.

1) The reviewer suggests changing the range of the Y-axis in figure 8c to match the other presented results of separation efficiency.

The Y-axis in fig. 8c has been changed according to the reviewer's suggestion

2) Furthermore, the authors should check the spelling and grammar in added text.

The added text was fully revised, and spelling and grammar were corrected

Reviewer 3 Report

The author has made full revisions to the review comments.

Author Response

Thanks for the helpful comments and suggestions.

Reviewer 4 Report

The authors made all necessary modifications. Therefore, the manuscript can be accepted for publication.

Author Response

(The authors gave the same response as above.)
